# Investigating the spatiotemporal differences and influencing factors of green water use efficiency of Yangtze River Economic Belt in China

**Ke-Liang Wang**[1], **Jianguo Wang**[2]*, **Jianming Wang**[2], **Lili Ding**[1], **Mingsong Zhao**[3], **Qunwei Wang**[4]

1 School of Economics, Ocean University of China, Qingdao, PR China, 2 School of Business Administration, Zhejiang University of Finance and Economics, Hangzhou, PR China, 3 School of Surveying and Mapping, Anhui University of Science and Technology, Huainan, PR China, 4 School of Economics and Management, Nanjing University of Aeronautics and Astronautics, Nanjing, PR China

* cancou1005@163.com

**Data Availability Statement:** All relevant data are within the manuscript and its Supporting Information file.

## Abstract

Combining freshwater consumption and wastewater emissions into a unified analysis framework and utilizing the epsilon-based measure (EBM) model with the characteristics of radial model and non-radial model, this paper evaluates green water use efficiency (GWUE) of 11 provincial-regions in the Yangtze River Economic Belt (YREB) and investigates its spatiotemporal differences during the period 2005–2014, on basis of which the contribution rate of each input-specific green water use inefficiency in the overall green water use efficiency and the potential of freshwater-saving and wastewater emissions reduction are also calculated. The Theil index is used to explore the sources of the provincial gap of green water use inefficiency, and a random-effect panel Tobit model is applied to test the impact of the influencing factors of green water use inefficiency in the YREB. It is found that green water use inefficiency of the YREB is relatively low and regional differences is significant during the sample period, indicating a large potential of water-saving and water pollution reduction, and narrowing *BGAP* and *WGAP* of the Upstream is the key for improving green water use inefficiency in the YREB. The panel Tobit regression results show that economic development, technological innovation, water use structure, water resources endowment, environmental regulation and regional differences all play positive/negative effects on green water use inefficiency in the YREB, while these factors' influencing direction, degree and significance are significantly different. The conclusions of our study can provide considerably valuable information for the YREB to reserve water resources and reduce wastewater emissions.

## 1. Introduction

The Yangtze River is China's largest river, the third largest river in the world, flowing through nine provinces and two municipalities of China. The length of the main stream is 6,300

**Funding:** We gratefully acknowledge the financial support provided by the National Natural Science Foundation of China (Nos.71973131 and 71403003), Key Project of The National Social Science Fund of China (No.18AZD015), Major Project of The National Social Science Fund of China (No.19VHQ002), The China Postdoctoral Science Foundation (Nos. 2014M55187 and 2015T80643) and the Anhui Philosophy Social Science Planning Project (No. AHSKY2018D92).

**Competing interests:** NO authors have competing interests.

kilometers, and the basin covers an area of more than 1.8 million square kilometers, accounting for about one-fifth of China's land area. "The Yangtze River Economic Belt (YREB)" was first proposed in the 1980s, which originated from the "Yangtze River Industrial Dense Belt" proposed by the China Productivity Economics Research Association, which is an economic belt along Yangtze River with Shanghai as the leader. Promoting the development of the YREB has become a national strategy in 2015. As the world's largest river economic belt, the YREB accounts for more than 40% of China's population as well as gross domestic product (GDP), and has played a critical influence in the course of China's regional economic development [1]. However, long-term high-intensity economic development has caused the YREB to face severe resource and environmental problems, especially the water shortage and water pollution problems. On the one hand, the long-term extensive water use pattern leads to low water use efficiency and serious waste of water resources in the Yangtze River Basin. It is foreseeable that with the acceleration of industrialization and urbanization, the water resources demand will continue to increase in the future and become a key constraint factor for economic and social development in the YREB [2,3,4]. On the other hand, due to the lack of effective supervision from local government, a large amount of industrial wastewater and domestic sewage have not been treated and emissions directly into the Yangtze River, resulting in more and more serious water pollution. To sum up, water shortage, water pollution and poor water management have seriously threatened the sustainable development of YREB, and thus it is imperative to take powerful commitments to rectify this passive situation. In this context, improving water use efficiency and reducing wastewater emissions, have been recognized as two better ways to alleviate the ongoing water crisis and support sustainable development in the YREB [4,5]. Therefore, it is of considerable importance to scientifically measure the water use efficiency incorporating wastewater emissions (e.g. green water use efficiency) of different sub-regions in the YREB and investigate its influencing factors, which can help both scholars and policy-makers clarify the gains and losses in the process of water use, form a feedback mechanism for adjustments and improvement in follow-up policies, and determine the appropriate direction and focus of future work [6].

In recent years, water use efficiency has acquired increasing attention from academic researchers, and scholars have used different types of models to evaluate water use efficiency [7–15]. As a conventional method for measuring the relative efficiency of decision-making units (DMUs), the data envelopment analysis (DEA) has been widely used in the estimation of water use efficiency. DEA is a nonparametric method which was first proposed by Charnes et al. [16], and it does not require any restricting assumptions of the related function between multiple input and output variables, and it need not to obtain the price information of input and output factors, thus the estimation is easier and the efficiency evaluation would be more objective [17]. Hu et al. [18] first defined total-factor water efficiency index by applying conventional radial DEA for evaluating provincial total-factor water efficiency in China. So far, the total-factor framework has been widely utilized to investigate total-factor water efficiency in different countries and regions [19–26]. Generally, in the conventional total-factor water efficiency framework, three key input factors (water, capital and labor) as well as the economic output factors (GDP) are all included, which can reflect the substitution between different input factors in the production process. However, the above conventional water efficiency evaluations only considered desirable outputs (e.g. economic outputs) and simply ignored undesirable outputs (e.g. wastewater emissions). Therefore, the social welfare and economic performance in the process of water use activity is distorted. In recent years, progressively greater numbers of scholars have realized the shortcoming, and tried to incorporate wastewater emissions into the water use efficiency evaluation framework. Li and Ma [3] used SBM-undesirable and meta-frontier models to evaluate China's industrial water use efficiency and

investigated the impact factors. Wang et al. [4] explored agricultural water-use efficiency in the Heihe River Basin in the Northwest China with DEA and Malmquist productivity index. Deng et al. [6] employed SBM-DEA model to evaluate water use efficiency of 31 provinces in China with the sewage as undesirable output. Zhao et al. [5] adopted two-stage SBM model to estimate water resource utilization efficiency with taking environmental constraint into account and investigated the spatial spillover effect. Song et al. [17] applied Malmquist-Luenberger productivity index to measure provincial water resource efficiency in China and used panel Tobit model to explore the factors that affect water resource efficiency. Yao et al. [27] measured green total factor water efficiency by using SBM model considering undesirable output. Hu et al. [28] utilized a super efficiency DEA to evaluate the water use and wastewater treatment efficiency of 10 cities in the Minjiang River Basin of China. In these studies, the economic outputs as well as negative externalities generated (e.g. wastewater emissions) during water use activity are well considered. Therefore, water use efficiency evaluations considering wastewater emissions can provide more reasonable and more accurate estimation results.

Nevertheless, there are still some shortcomings in the existing studies related to water use efficiency. First, the existing studies used radial or non-radial DEA approaches to estimate water use efficiency. However, the main shortcoming of the radial models, represented by the conventional CCR and BCC model, is the neglect of non-radial slacks when estimating efficiency, which may lead to a biased estimation result. In addition, the radial models require input or output variables to change proportionally, which cannot cope with practical production process properly [29,30]. The non-radial models, represented by the slacks-based measure (SBM) model proposed by Tone [31], aim at obtaining the maximum rates of reduction in inputs, relaxing the proportionality constraint and allowing independent changes to associated slacks. However, in the SBM model, the projected DMU may lose proportionality with the original input, which is inappropriate for the analysis [32]. To sum up, both radial and non-radial measure models have defects, and thus may cause a certain deviation of the results. Secondly, most of the existing studies about water use efficiency utilize single-phase benchmark technologies to construct single-phase reference production sets. In this way, only the efficiency difference of different DMUs at the one period can be considered, while the efficiency change trend of the same DMU at different periods cannot be obtained. As a result, estimation results from these studies is incomplete and cannot provide more detailed information about water use efficiency [33–40].

In order to overcome the above shortcomings remaining in the existing studies, we carry out water use efficiency evaluation by applying a novel DEA approach—epsilon-based measure (EBM) model considering wastewater emissions in our analysis. The EBM model, first proposed by Tone and Tsutsui [29], which compiles the radial model and non-radial model into a composite model to measure efficiency in a more reasonable way, and effectively addresses the weaknesses of the radial and non-radial models. In recent years, the EBM model has been widely applied in the field of evaluation of efficiency and productivity. Qin et al. [32] employed global EBM model to estimate the energy efficiency with air emission as undesirable output in China's coastal areas. Xu and Cui [41] applied a new approach combining network EBM model and network SBM model to evaluate the overall energy efficiency and divisional efficiency of 19 international airlines. Cui and Li [42] used a dynamic EBM model to evaluate the dynamic efficiency of 19 international airlines. The above studies show that the EBM model is feasible and effective for measuring the efficiency of product units with the advantages of the radial and non-radial models simultaneously. However, to the best of our knowledge, studies that use the EBM model to measure water use efficiency have not been found. To fill this gap, in this paper, we utilize a window-based EBM model to investigate the water use efficiency of the YERB at a more detailed level. Major contributions of this study can be

summarized as follows. First, it combines the EBM model and window-based DEA approach to evaluate and decompose provincial green water use efficiency in the YREB. Second, the Theil index is employed to identify the sources of the gap of green water use efficiency among the three major areas of the YREB. Third, a random-effect panel Tobit model is used to test the influencing factors of green water use efficiency in the YREB.

The remainder of this paper is structured as follows: Section 2 introduces the definition of green water use efficiency, the EBM model, The Theil index and the panel Tobit regression model. Section 3 describes the data source and variables selection. In Section 4, the above proposed approaches are used to systematically examine the spatial-temporal differences and influencing factors of green water use efficiency of the YREB in China. Conclusions are presented in Section 5.

## 2. Methodology

In this section, we present the models and analysis approaches to investigate the spatiotemporal differences and influencing factors of green water use efficiency of the YREB in China. As the model and analysis involve various variables and abbreviations, we first provide a summary of the abbreviations and notations used in this study (see Table 1).

### 2.1 Green water use efficiency (GWUE)

Water use efficiency, defined by the ratio of the optimal water input to the actual water input of an economic unit, is an important indicator for measuring the level of water resource utilization for a country, region or firm. However, most of the existing water use efficiency related studies mainly focused on the water conservation but did not consider the discharge of wastewater emissions, and thus the results were biased. To address this problem, this paper incorporates wastewater emissions into the total-factor water use efficiency measurement framework

**Table 1. Summary of the abbreviations and notations with their descriptions.**

| Notation and abbreviation | Description |
|---|---|
| YREB | Yangtze River Economic Belt |
| GWUE | Green water use efficiency |
| EBM | Epsilon-based measure |
| DMU | Decision-making unit |
| CRS | Constant returns to scale |
| VRS | Variable returns to scale |
| SBM | Slack-based measure |
| PTPS | Production technology possible set |
| TE | Technical efficiency |
| PTE | Pure technical efficiency |
| SE | Scale efficiency |
| ISGWUI | Input-specific green water use inefficiency |
| ISGWUE | Input-specific green water use efficiency |
| CON | Contribution rate of each input-specific green water use inefficiency to the overall green water use inefficiency |
| OGAP | Overall provincial green water use efficiency gap |
| BGAP | Green water use efficiency gap between three major areas in the YREB |
| WGAP | Provincial green water use efficiency gap within the three major areas in the YREB |
| MGAP | Provincial green water use efficiency gap of the specific three major areas in the YREB |

and defines a new water use efficiency indicator named "green water use efficiency (GWUE)" by utilizing a novel DEA model—EBM model. The new indicator takes into account both freshwater conservation and wastewater mitigation simultaneously, thereby providing a more scientific and comprehensive approach for water use efficiency measurement.

According to Kuosmanen and Kortelainen [43], Picazo-Tadeo et al. [44], in this paper we define green water use efficiency as a ratio between economic value added and water environmental pressures. Assume that the economic value added, denoted by variable $v$, generated in the water use processes by a set of $k = 1,...,K$ provinces. In addition, the water use process generates a set of $i = 1, ..., m$ damaging water environmental pressures including freshwater consumption and wastewater emissions, which are denoted by $p = (p_1, p_2,..., p_m)$. The production technology possible set (*PTPS*) representing all feasible combinations of economic value added $v$ and water environmental pressures $p$ is defined as follows.

$$PTPS = \{(v, p)| \text{value added } v \text{ can be generated with water environmental pressures } p\} \quad (1)$$

Following Picazo-Tadeo et al. [44], the green water use inefficiency (GWUE) of province $k = 1,...,K$ is defined as follows,

$$GWUE_k = v_k/P(p_k) \quad (2)$$

Where $P$ is the water environmental pressures function that aggregates the $m$ water environmental pressures into a single water environmental pressure index. Following Kuosmanen and Kortelainen [43] and Picazo-Tadeo et al. [44], we considered that a reasonable approach for the calculation of water environmental pressures function is to take a weighted average of pressures exercised by province $k$ ($k = 1,..,K$). As a result, the water environmental pressures index can be calculated as follows.

$$P(p_k) = \sum_{i=1}^{m} w_i p_{ik} \quad (3)$$

Where $w_i$ ($i = 1,...,m$) is the weight of water environmental pressure $i$ ($i = 1,...,m$) for the computation of environmental pressures function. Since $w_i$ ($i = 1,...,m$) represents the relative importance of different environmental pressures, it is extremely essential to utilize which method to accurately obtain the weight of each water environmental pressure for the evaluation of green water use inefficiency. Aiming to avoiding the bias due to subjective choice of traditional weights, in this paper we choose DEA approach as the aggregation method [44].

## 2.2 Epsilon-based measure (EBM) model

As noted in the Section 1, both the radial and non-radial models suffer from some shortcomings. To overcome these shortcomings, Tone and Tsutsui [29] proposed the "epsilon-based measure (EBM)" model by combing both these two types of models into a unified framework. In recent years, the EBM model has been widely applied in the field of energy and environmental efficiency evaluation and shows that this model can effectively addresses the defects of the radial and non-radial models [30,32,41,42]. As a result, we utilize the EBM model to evaluate the green water use efficiency for each province in the YREB in China. Formally, the input-

oriented EBM model which exhibits constant returns to scale (CRS) is expressed as follows.

$$\min\gamma = \theta - \varepsilon_p \sum_{i=1}^{m} \frac{w_i^- s_i^-}{p_{i0}}$$

$$s.t.\ \theta p_{i0} = \sum_{k=1}^{K} z_k p_{ik} + s_i^-; v_0 \leq \sum_{k=1}^{K} z_k v_k; z_k \geq 0, s_i^- \geq 0, i = 1, 2, ..., m$$

(4)

Where the subscript "0" represents the given province under evaluated, $\gamma$ is green water use efficiency of the given province ranging from 0 to 1, when the efficiency score is equal to 1, it indicates that the given province is on the frontier. $s_i^-$ denotes the input slacks of the $i$th input, $z_k$ is the intensity variable, $w_i^-$ is the weight of the $i$th input variable, and it is satisfied with $\sum_{i=1}^{m} w_i^- = 1(w_i^- \geq 0, \forall i)$. $\varepsilon_p$ is a key parameter which combines the terms of the radial and non-radial slacks. It can be found that the EBM model will be simplified to the input-oriented CCR model when $\varepsilon_p = 0$, but it will be the input-oriented SBM model when $\varepsilon_p = 1$. The detailed information about its calculation can be obtained in Tone and Tsutsui [29]. It should be noted that the efficiency evaluation of formula (4) is under constant returns to scale (CRS) assumption. If we add the constraint of $\sum_{k=1}^{K} z_k = 1$, that the efficiency evaluation is under variable returns to scale (VRS) assumption. By calculating the two formulas, the pure technical efficiency (PTE) and scale efficiency (SE) of technical efficiency (TE) of water resources utilization can be obtained, and TE = PTE×SE. It should be highlighted that, according to the definition of green water use efficiency by formula (2), all water environmental pressures including freshwater consumption and wastewater emissions, are all regarded as inputs in our study. This treatment of water environmental pressures is commonly used and has been widely applied for the evaluation of eco-efficiency and environmental performance in the past few years [43–49]. That's to say, in this paper, wastewater emissions are not considered as an undesirable output along with economic output, but are deemed as environmental costs in the water use process, and thus input factors.

In order to identify the sources of green water use inefficiency, we further decompose green water use inefficiency from the perspective of each water environmental pressure. Since $w_i^- (i = 1, ..., m)$ is the weight of $i$th water environmental pressure, the objective function $\gamma$ of the EBM model can be decomposed as follows.

$$\gamma = \sum_{i=1}^{m} w_i^- (\theta - \varepsilon_p s_i^- / p_{i0})(i = 1, ..., m)$$

(5)

The value of green water use inefficiency for each water environmental pressure, namely input-specific green water use inefficiency (ISGWUI), can be expressed as follows.

$$ISGWUI_i = 1 - (\theta - \varepsilon_p s_i^- / p_{i0})(i = 1, ..., m)$$

(6)

According to formula (6), the value of green water use efficiency for each water environmental pressure called input-specific green water use efficiency (ISGWUE) can be defined as follows.

$$ISGWUE_i = \theta - \varepsilon_p s_i^- / p_{i0}(i = 1, ..., m)$$

(7)

Based on formulas (5) and (6), the contribution rate of each input-specific green water use inefficiency to the overall green water use inefficiency (CON) can be calculated by the

following formula.

$$CON_i = (w_i^- \times ISGWUI_i)/(1 - \gamma) \quad (i = 1, ..., m) \tag{8}$$

## 2.3 Theil index and its decomposition

The Theil index is a weighted entropy index which was proposed by Theil [50] and originally used to measure economic inequality. As a special form of the generalized entropy index system, in this paper, the Theil index is used to measure the provincial green water use efficiency gap (*OGAP*)and decompose it into two components, namely the green water use efficiency gap between three major areas (*BGAP*) and the provincial green water use efficiency gap within the three major areas in the YREB (*WGAP*). As a result, the changing trend and the sources of the provincial green water use efficiency gap in the YREB can be obtained (In order to explore the sources of provincial green water use efficiency gap in the YREB, we divide its 11 provincial-regions into three major areas according to the geographical location, namely the Upstream, the Midstream and the Downstream respectively, and it will be further discussed in the following section). The Theil index value ranges from 0 to 1. The smaller the value, then the smaller the provincial gap and vice versa. Accordingly, the Theil index and its decomposition components of green water use efficiency in the YREB can be formulated as follows.

$$OGAP = \sum_{i=1}^{11} (e_i/\bar{e})\ln(e_i/\bar{e})/11 \tag{9}$$

$$MGAP_h = \sum_{i=1}^{n_h} (e_{hi}/\bar{e}_h)\ln(e_{hi}/\bar{e}_h)/n_h \tag{10}$$

$$WGAP = \sum_{h=1}^{3} (n_h\bar{e}_h/\bar{e})MGAP_h/11 \tag{11}$$

$$BGAP = \sum_{h=1}^{3} n_h(\bar{e}_h/\bar{e})\ln(\bar{e}_h/\bar{e})/11 \tag{12}$$

$$OGAP = WGAP + BGAP \tag{13}$$

Where $n_h$ ($h$ = 1,2,3) denotes the number of provincial-regions in the three major areas of the YREB in China. $e_i$ ($i$ = 1,. . .,11) represents the green water use inefficiency of each provincial-region in the YERB. $\bar{e}$ denotes green water use efficiency average value of the 11 provincial-regions in the YERB. $\bar{e}_h$ ($h$ = 1,2,3) denotes green water use efficiency average value of each major area, respectively. $\bar{e}_{hi}$ ($h$ = 1,2,3;$i$ = 1,. . .,$n_h$) represents green water use efficiency of each provincial-region in the YREB's three major areas. $MGAP_h$ ($h$ = 1,2,3) represents the Theil index of the provincial green water use inefficiency gap of the three major areas.

## 2.4 Tobit model

To investigate the influencing factors of green water use efficiency in the YREB, econometric model is required in this paper. In order to select suitable econometric models, it is critical to understand the characteristic of estimated dependent variables. From Section 2.2, it can be obtained that dependent variables in our study, which includes the overall and input-specific

green water use efficiencies, are all between 0 and 1. According to McDonald and Moffitt [51], if dependent variables are truncated or censored, the Tobit model proposed by Tobin [52] is the proper choice. Therefore, in this paper, we choose the Tobit model for exploring the influencing factors of overall and input-specific green water use inefficiencies in the YREB. The Tobit model can be defined as follows.

$$y_i = \begin{cases} X_i\beta_i + u_i \ if \ y_i \in (0,1] \\ 0 \quad if \ y_i \in (-\infty, 0] \\ 1 \quad if \ y_i \in (1, +\infty) \\ i = 1, 2, ..., N \end{cases} \tag{14}$$

Where $N$ is the number of observations; $y_i$ is dependent variable; $X_i$ is the vector of independent variables; $u_i$ is the independently distributed error term and is assumed to be normal with zero mean and constant variance $\delta^2$.

## 3. Data source and variables selection

Based on the data availability, in this paper we consider the YREB's 11 provincial-regions as the sample for the period between 2005 and 2014. As pointed out in the above section, in order to investigate the regional differences of green water use inefficiency, the YREB is divided into three major areas, namely the Upstream, Midstream and Downstream, which is shown in Fig 1. Specifically, the Downstream locates on the coast of China and is constituted of two provinces and one municipality (Jiangsu, Zhejiang and Shanghai). This area is the most developed area in the YREB and is also one of the most developed area in China. The Midstream consists of four central provinces (Anhui, Jiangxi, Hunan and Hubei) of China. This area is China's energy and manufacturing bases and owns a large amount of water and environmental-intensive industries, which leads to a huge water consumption and related water pollutant emissions. The Upstream includes three western provinces and one municipality (Sichuan, Yunnan, Guizhou and Chongqing) of China. In comparison with the other two areas, this area has a lower population density and is the relatively backward area in the YREB.

As noted in Section 2, according to Kuosmanen and Kortelainen [43], Picazo-Tadeo et al. [44], in this paper green water use efficiency is defined as the ratio of the water environmental pressures (e.g. freshwater consumption and wastewater emissions) to economic value added, while the traditional production inputs such as capital and labor are not included. Although this definition violates the features of the practical production process to a certain extent, it can more accurately represent the trade-off between water resources utilization, water environmental protection and economic development with eliminating the influences of the utilization efficiencies of other conventional inputs (such as capital, labor, etc.), rather than a comprehensive utilization efficiency index of all input factors. From this perspective, the definition of green water use inefficiency in this paper could be more reasonable. Accordingly, in this paper, freshwater consumption and wastewater emissions are all considered as inputs, and economic value added as output for evaluating green water use inefficiency of the YERB in China.

More concretely, as described above, water environmental pressures including freshwater consumption and wastewater emissions are chosen as inputs in this study, including industrial freshwater consumption ($p_1$), agricultural freshwater consumption ($p_2$) and domestic freshwater consumption ($p_3$), industrial wastewater emissions ($p_4$) and urban domestic sewage emissions ($p_5$). The inputs data is collected from the 'China Environmental Statistical Yearbook' (2005–2015), 'China Statistical Yearbook' (2005–2015). For the output variable, regional Gross

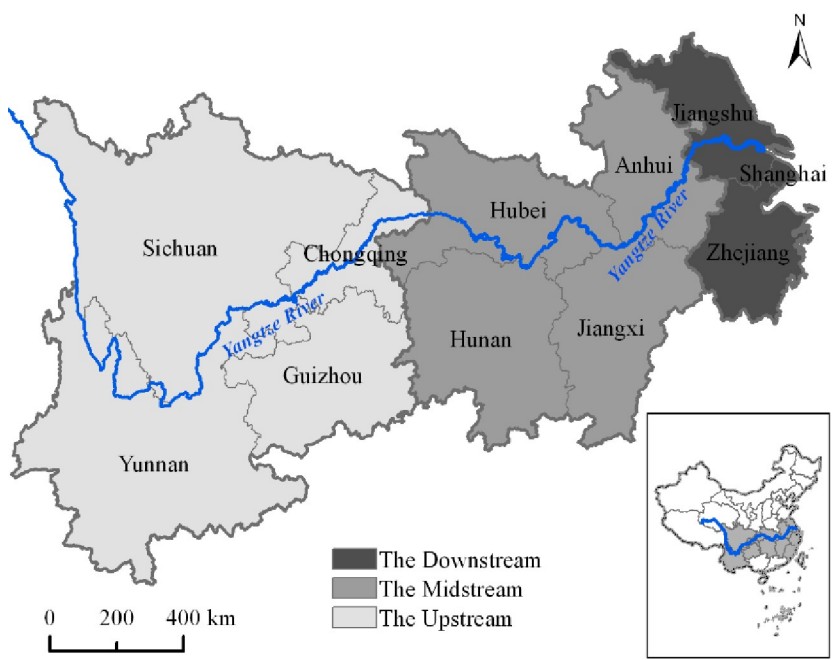

**Fig 1. The administrative regions and three major areas in the YREB.**

domestic product (GDP) is regarded as the most important indicator to reflect the economic performance for a region. Although the indicator has certain limitations for calculating economic value added, it is still a relatively good index to evaluate the output of economic activity due to its easy and mature computation method [53,17]. As a result, regional GDP ($v$) is chosen as output variable in this paper, of which data is also taken from the 'China Statistical Yearbook' (2005–2015) and converted into 2005 constant prices. The descriptive statistics of the inputs and outputs, at the provincial level of YREB in China, are given in detail in Table 2.

## 4. Empirical results and discussion

In this section, in order to obtain more detailed information, we utilize DEA window analysis as the efficiency estimation approach. DEA window analysis approach is a variation of traditional DEA approach that establishes efficiency estimations by considering each DMU in a different period as a separate DMU, and the efficiency of a DMU in a period can be contrasted with its own efficiency in other periods as well as to the efficiency of other DMUs, and thus the efficiency changing trends of the estimating DMU can be explored through a sequence of overlapping windows [30,54,55]. In addition, window analysis approach also can be used to

**Table 2. Descriptive statistics of inputs and outputs at provincial level.**

| Index | Observations | Unit | Mean | Stdev | Maximum | Minimum |
|---|---|---|---|---|---|---|
| $p_1$ | 121 | 100-million m$^3$ | 121.96 | 76.45 | 305.35 | 14.60 |
| $p_2$ | 121 | 100-million m$^3$ | 75.03 | 48.50 | 238.00 | 17.87 |
| $p_3$ | 121 | 100-million m$^3$ | 29.88 | 10.77 | 52.91 | 13.14 |
| $p_4$ | 121 | 10$^4$ t | 93053.28 | 68237.93 | 296318.00 | 11695.00 |
| $p_5$ | 121 | 10$^4$ t | 145358.61 | 75200.86 | 395931.00 | 39568.00 |
| $v$ | 121 | 10$^5$ thousand Yuan | 11928.23 | 8935.74 | 48410.63 | 1591.90 |

effectively reduce the lower efficiency discrimination problem coming from a relatively small number of DMUs [56]. To sum up, by window analysis approach, an analysis of green water use efficiency from a spatiotemporal perspective can be presented. It should be noted that windows analysis approach assumes that there are no technical changes with each window, which is regarded as a general problem of this approach [54,57]. To reduce this problem, in this paper we select a relatively narrow window length of two to get more credible efficiency results, and thereby the reference technique set of each year is constituted by the value of input and output of the current year and the last year. For this reason, the initial study year of this paper is changed from 2004 to 2005.

## 4.1 Analysis of spatiotemporal differences of green water use efficiency

Based on the provincial panel data, we obtained green water use inefficiency and its decomposition components of the YREB's 11 provincial-regions and three major areas, as shown in Table 3, Figs 2 and 3, respectively. It can be referred that the mean value of green water use inefficiency in the YREB was as high as 0.7826 during the period 2006–2014. This means that both freshwater consumption and wastewater emissions were not effective, and the potential of freshwater conservation and wastewater emissions abatement was relatively large in the YREB. According to the decomposition result of green water use inefficiency, the mean values of *PTE* and *SE* were respectively 0.8901 and 0.8791. This shows that both *PTE* and *SE* still have considerable improvement potentials and *PTE* is slightly higher than *SE*. As a result, in order to further upgrade the overall green water use inefficiency, in the future work, while improving the water resources utilization technology and management level to acquire a higher *PTE*, all regions in the YREB also should require to properly control the total amount of water consumption and wastewater emissions for promoting *SE*.

As shown in Table 3 and Fig 2, at the provincial level, Shanghai and Zhejiang had always been at the technological frontier with the mean green water use inefficiency value of 1 during the sample period. This indicates that their freshwater consumption and wastewater emissions are optimal relative to other provincial-regions in the YREB. Green water use efficiency of Jiangsu, Sichuan and Yunnan were all above 0.80, while the green water use inefficiency values of Anhui, Jiangxi, Hunan and Guizhou were relatively low, all below 0.70, and the least-efficient region was Hunan with the value of 0.6293. This suggests that these four regions have serious water waste and water pollution problem and it is considerably imperative to improve

**Table 3. The mean value of green water use efficiency and its components in the YREB during the sample period.**

| Provincial-region | TE | PTE | SE |
|---|---|---|---|
| Shanghai | 1.0000 | 1.0000 | 1.0000 |
| Jiangsu | 0.8913 | 1.0000 | 0.8913 |
| Zhejiang | 1.0000 | 1.0000 | 1.0000 |
| Anhui | 0.6902 | 0.7452 | 0.9261 |
| Jiangxi | 0.6352 | 0.7715 | 0.8234 |
| Hubei | 0.7284 | 0.7499 | 0.9713 |
| Hunan | 0.6293 | 0.6541 | 0.9621 |
| Chongqing | 0.7224 | 0.9991 | 0.7231 |
| Sichuan | 0.8484 | 0.8772 | 0.9672 |
| Guizhou | 0.6593 | 0.9958 | 0.6621 |
| Yunnan | 0.8047 | 0.9985 | 0.8060 |
| Mean value | 0.7826 | 0.8901 | 0.8791 |

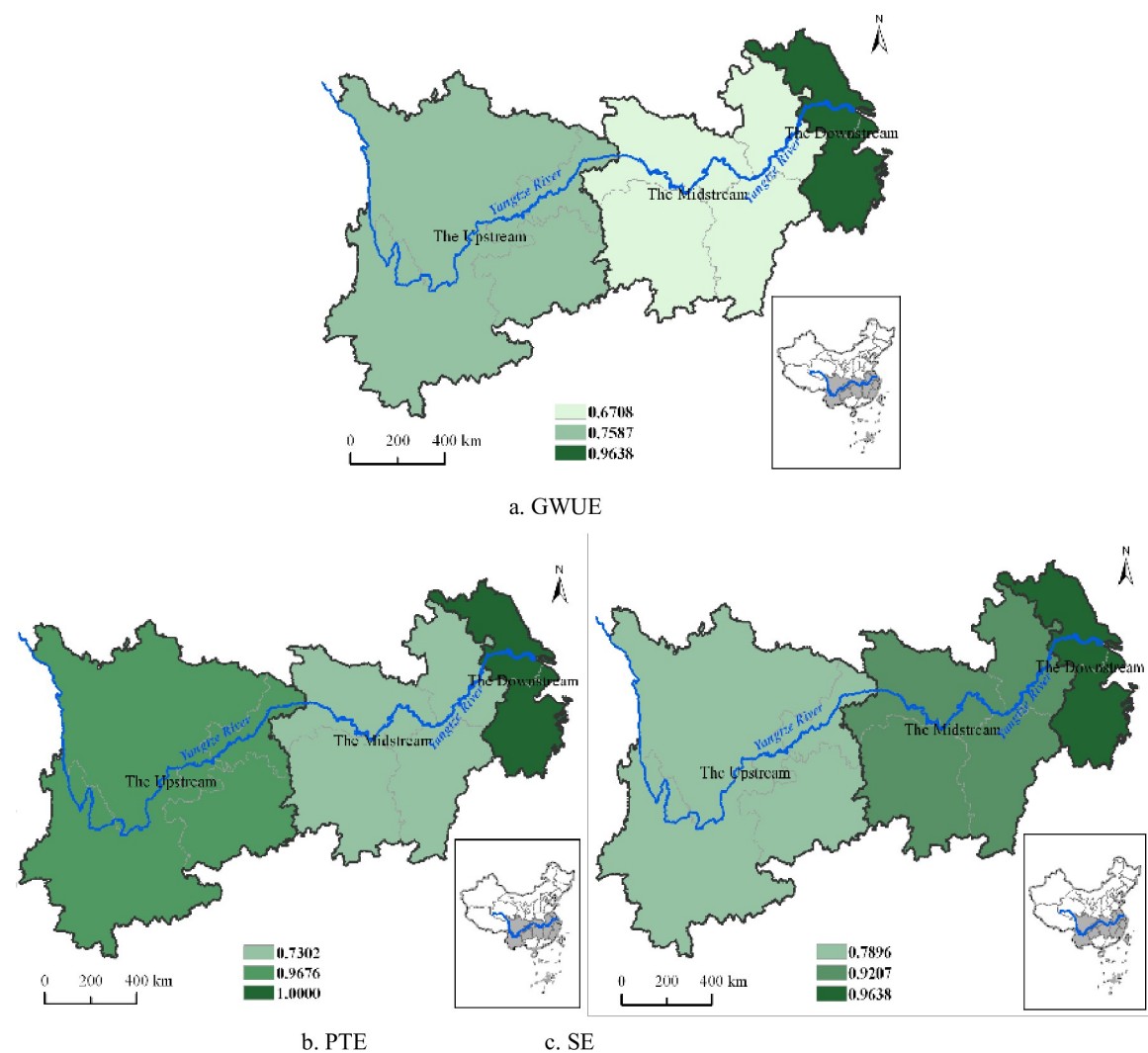

a. GWUE

b. PTE          c. SE

**Fig 2. The distribution of green water use efficiency and its components in the YREB.**

the management level of water resources conservation and water environmental protection in these regions for realizing the sustainable development of water resources.

It can be seen from Table 3 and Fig 2, the *PTE* of Shanghai, Jiangsu and Zhejiang from the Downstream was the highest with the score of 1, showing their the most advanced technology and management level in the domain of water use in the YREB. The *PTE* scores of Chongqing, Guizhou and Yunnan from the Upstream were all above 0.90, showing the fact that the Upstream water-saving society construction had achieved good results in recent years. In contrast, the *PTE* scores of the four regions from the Midstream were at relatively low level, all below 0.70. There are several reasons for this. First, the Midstream is China's main rice-growing area, and its agricultural freshwater consumption is very huge. Second, the Midstream is also one of the most densely populated areas in China, thereby the domestic freshwater consumption and urban domestic sewage emissions are also significantly higher than the other two major areas in the YREB. The third is that the Midstream has long been China's energy and manufacturing base with high industrial freshwater consumption and wastewater emissions, facing serious problems of water conservation and water environmental protection.

a. GWUE

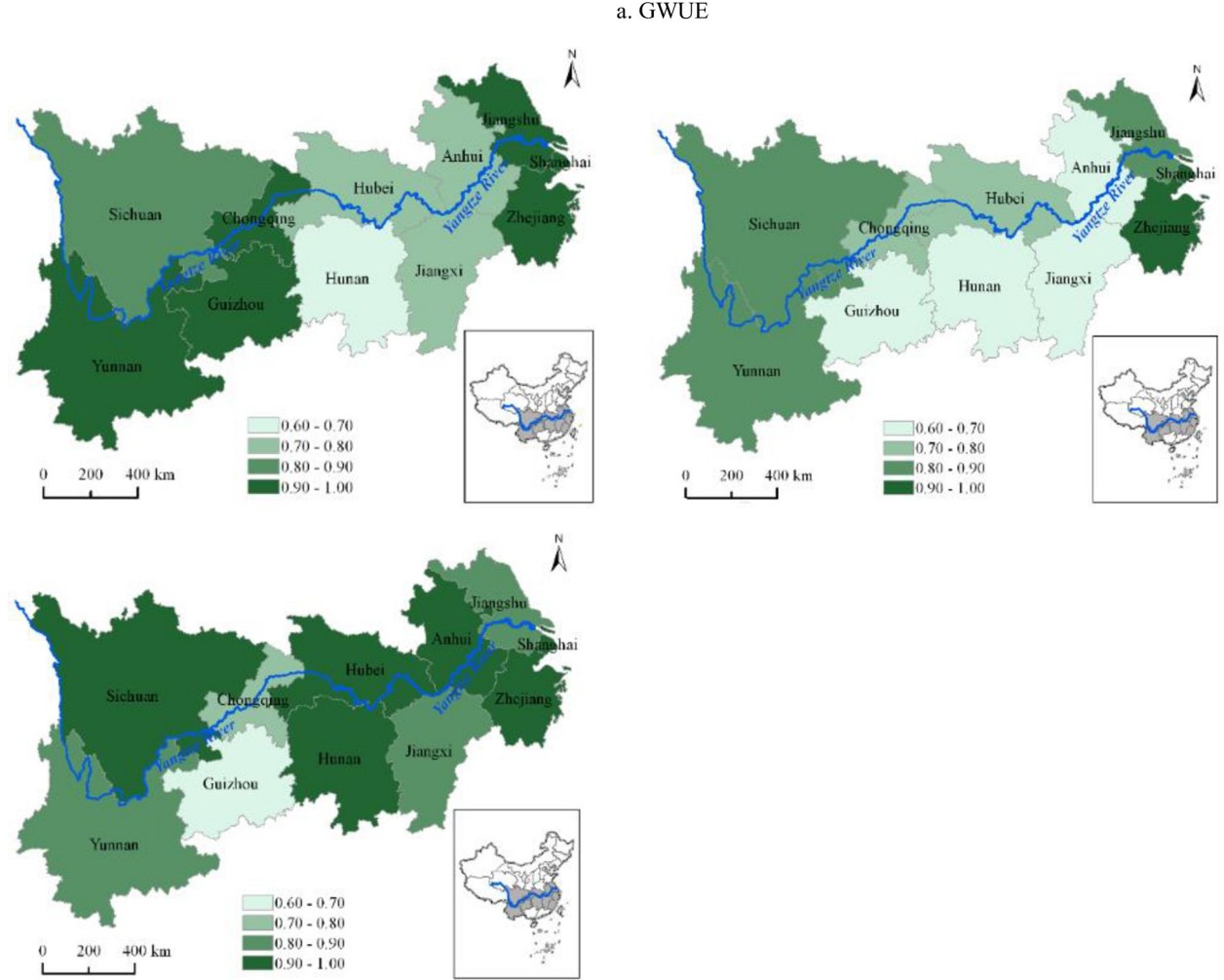

**Fig 3. The distribution of green water use efficiency and its components of the three major areas in the YREB.**

What's more, large amount of resource and environmental-intensive manufacturing industries in the Downstream continuously moved to the Midstream in recent years, leading to a rapid increase of freshwater consumption and wastewater emissions in the Midstream, which severely restricts the further improvement of the Midstream green water use efficiency.

Regarding *SE*, as shown in Table 3 and Fig 2, the most-efficient region was Shanghai and Zhejiang with the *SE* value of 1. This shows that the freshwater consumption and wastewater emissions scale of the two regions is relatively optimal compared to other regions in the YREB. Anhui, Hubei, Hunan and Sichuan also displayed a relatively higher *SE* with the score more than 0.90. It should be noted that, as the least-efficient region in the Downstream, Jiangsu's green water use inefficiency was mainly hampered by its lower *SE*. This is mainly due to the excessive of water inputs, resulting in diminishing returns to scale. This is supported by the original statistic data that Jiangsu was the highest in the 11 provincial-regions of the YREB, both in terms of the total freshwater consumption and wastewater emissions. According to the law of diminishing marginal returns, excessive inputs will inevitably lead to a decline in scale returns and ultimately prevent the resources efficiency improvement. In addition, it can be inferred that the lower *SE* hindered the improvement of green water use inefficiency in

Chongqing, Guizhou, Yunnan and Jiangxi. However, unlike Jiangsu, these regions were still in the stage of increasing returns to scale, and thus it is also possible to promote green water use inefficiency by increasing the scale of water resources inputs to some extent.

In terms of regional differences in the three major areas, it can be seen in Fig 3, the distribution of green water use inefficiency is uneven. Specifically, green water use inefficiency, *PTE* and *SE* in the Downstream were all the highest in the three major areas of the YREB during the sample period, reaching 0.9638, 1.0000 and 0.9638, respectively. Among them, the *PTE* was completely effective, and the green water use inefficiency was mainly due to the lower *SE*. However, the green water use inefficiency and *PTE* of the Midstream were the lowest among the three major areas with the values of 0.6708 and 0.7302, respectively, the main reason of which is the Midstream lower *PTE* originating from its backward water utilization technology and poor water resources management. The green water use inefficiency of the Upstream was mainly constrained by the lower SE resulting from the smaller scale of water resources inputs.

To make the estimation results of different years comparable, the concept of window analysis approach is introduced into our study. The changing trends of green water use inefficiency, *PTE* and *SE* in the YREB during the sample period are presented in Fig 4. It can be seen that the green water use inefficiency increased from 0.7486 in 2005 to 0.7769 in 2014, and showed a changing trend of invert the U-shape curve, and the peak appeared in 2010. The changing trends of *PTE* and *SE* were consistent with green water use inefficiency, their peaks also appeared in 2010 and then fell. To understand the results, it is necessary to be familiar with the background. Based on the analysis of the original statistics of the YREB's freshwater consumption and total wastewater emissions, it can be found that during the sample period, the total amount of agricultural, industrial, domestic freshwater consumption and the industrial wastewater emissions in the YREB all increased less, but urban domestic sewage emissions increased rapidly, more than the doubled during 2005–2014, and the increase rate accelerated significantly after 2010, which may be a possible cause of the declines of green water use inefficiency and its components in the YREB. Therefore, in order to promote green water use inefficiency, it is considerably necessary for the local governments in the YREB to conduct stricter control of urban domestic sewage emissions as quickly as possible.

At the provincial level, it can be inferred from Fig 5, during the sample period, except for Shanghai and Zhejiang with the optimal efficiency, green water use inefficiency in most other regions exhibited an increasing trend, only Jiangsu and Jiangxi displayed a decreasing trend. Furthermore, it also can be found that the peak efficiency of most regions appeared in 2010, which was in line with the overall YREB. From the three major areas prospect, it can be seen from Fig 6, green water use inefficiency of the three major areas exhibited similar changing trends during the sample period, and moreover, they all reached the peaks in 2010, which was also consistent with the overall YREB. It is also evident, based on Fig 6, the Upstream increased faster than the Midstream and Downstream, showing the achievement of the Upstream in water conservation and water environmental protection in recent years.

## 4.2 Analysis of the sources of provincial gap of green water use efficiency

Identifying the sources of provincial green water use inefficiency gap in the YREB is conducive to providing more detailed information for policy-making. Therefore, in this study, the Theil index is introduced to investigate the sources of provincial green water use inefficiency gap in the YREB. The calculation results and its components of the Theil index are displayed in Table 4.

It can be found in Table 4 that, the Theil index of green water use inefficiency in the YREB showed an increasing trend during the sample period, from 0.0184 in 2005 to 0.0202 in 2014,

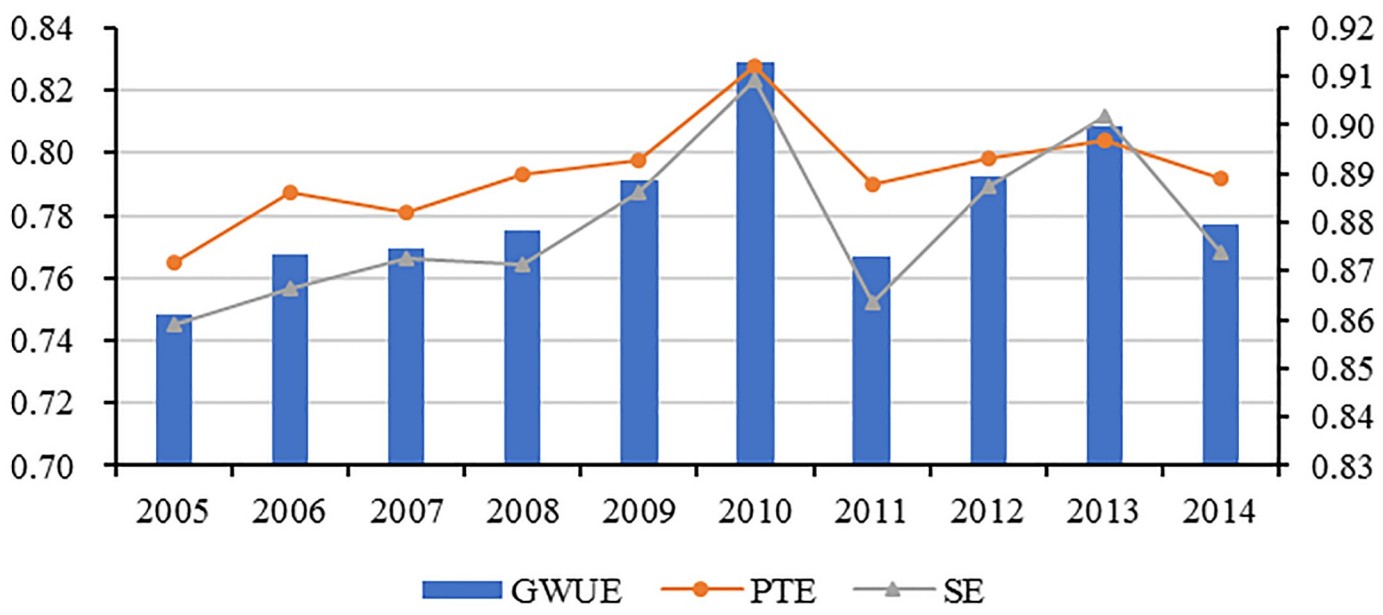

**Fig 4. The changing trend of green water use efficiency and its components of the YREB.**

suggesting that the provincial green water use inefficiency gap among the 11 provincial-regions was expanding, especially after 2010, the gap widened more significantly. According to

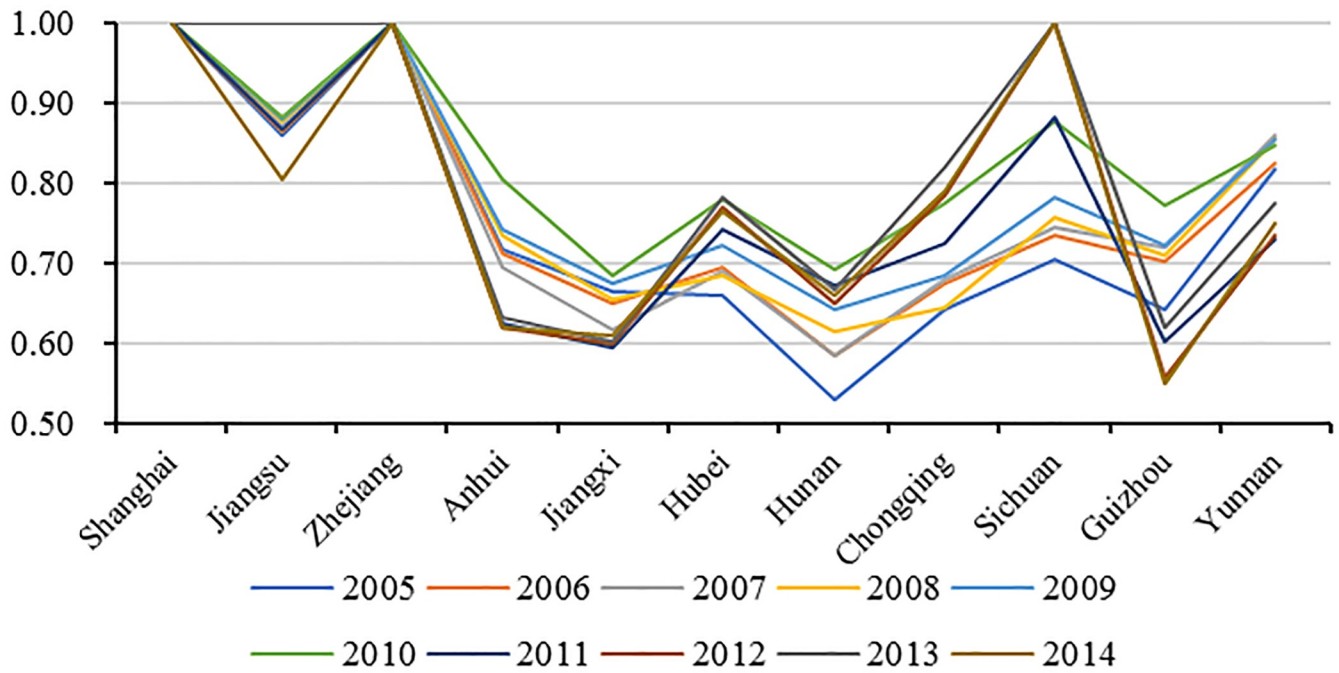

**Fig 5. The changing trend of green water use efficiency of 11 regions in the YREB.**

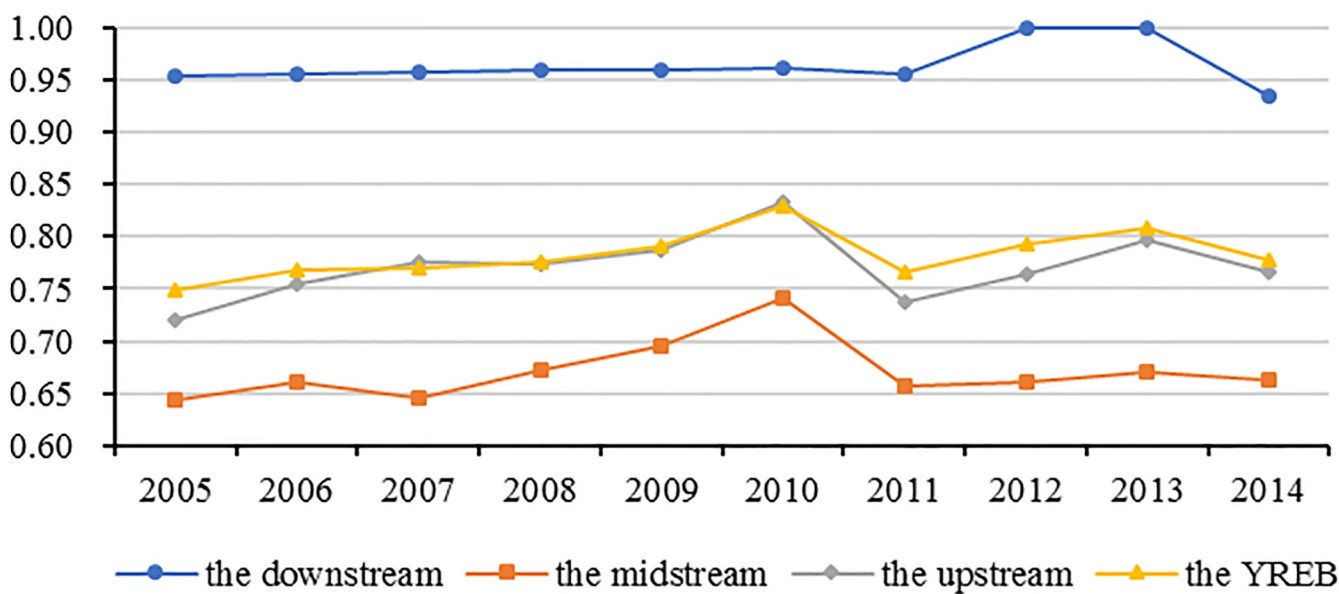

**Fig 6. The changing trend of green water use efficiency in the YREB and its three major areas.**

the decomposition results of the Theil index, it can be inferred that the contribution rate of *BGAP* was higher, reaching 70.29%, while the contribution rate of *WGAP* was only 29.71%. This indicates that *BGAP* was the main source of the overall provincial green water use inefficiency gap in the YREB. From the dynamic trend perspective, during 2005–2014, the contribution rate of *BGAP* dropped from 75.60% to 46.61%. By contrast, the contribution rate of *WGAP* increased from 24.40% in 2005 to 53.59% in 2014. This shows that the contribution rates of the above two types of gaps are both considerably large and should be narrowed simultaneously to improve the overall green water use inefficiency in the YREB. From the perspective of *WGAP*, the Upstream exhibited the largest contribution rate with an average of 18.65%, and its contribution rate increased over the sample period, from 9.57% in 2005 to 38.94% in 2014; the contribution rate of the Downstream was the smallest with an average of 4.25%; the

**Table 4. The Theil index and its components of green water use efficiency in the YREB.**

| Year | OGAP | WGAP | | BGAP | | The Downstream | The Midstream | The Upstream |
|------|------|-------|-------------------|-------|-------------------|-------------------|-------------------|-------------------|
| | | value | Contribution rate | value | Contribution rate | Contribution rate | Contribution rate | Contribution rate |
| 2005 | 0.0184 | 0.0139 | 75.60% | 0.0045 | 24.40% | 4.72% | 10.11% | 9.57% |
| 2006 | 0.0143 | 0.0116 | 81.55% | 0.0026 | 18.45% | 5.39% | 6.18% | 6.88% |
| 2007 | 0.0155 | 0.0127 | 81.48% | 0.0029 | 18.52% | 4.14% | 5.18% | 9.20% |
| 2008 | 0.0140 | 0.0109 | 77.55% | 0.0031 | 22.45% | 4.36% | 4.83% | 13.26% |
| 2009 | 0.0112 | 0.0089 | 79.11% | 0.0023 | 20.89% | 5.26% | 4.34% | 11.29% |
| 2010 | 0.0074 | 0.0055 | 74.18% | 0.0019 | 25.82% | 6.96% | 11.25% | 7.61% |
| 2011 | 0.0198 | 0.0119 | 59.78% | 0.0080 | 40.22% | 3.70% | 5.47% | 31.05% |
| 2012 | 0.0231 | 0.0142 | 61.46% | 0.0089 | 38.54% | 0.00% | 6.36% | 32.18% |
| 2013 | 0.0194 | 0.0127 | 65.62% | 0.0067 | 34.38% | 0.00% | 7.85% | 26.53% |
| 2014 | 0.0202 | 0.0094 | 46.61% | 0.0108 | 53.39% | 8.01% | 6.45% | 38.94% |
| Mean | 0.0163 | 0.0112 | 70.29% | 0.0052 | 29.71% | 4.25% | 6.80% | 18.65% |

**Table 5. Input-specific green water use inefficiency and its contribution rate.**

| Year | Input-specific green water use inefficiency | | | | | | Contribution rate | | | | |
|------|--------|--------|--------|--------|--------|--------|--------|--------|--------|--------|--------|
| | $p_1$ | $p_2$ | $p_3$ | $p_4$ | $p_5$ | total | $p_1$ | $p_2$ | $p_3$ | $p_4$ | $p_5$ |
| 2005 | 0.0741 | 0.0487 | 0.0579 | 0.0320 | 0.0386 | 0.2514 | 29.49% | 19.38% | 23.03% | 12.74% | 15.34% |
| 2006 | 0.0725 | 0.0472 | 0.0540 | 0.0287 | 0.0300 | 0.2324 | 31.19% | 20.33% | 23.23% | 12.33% | 12.91% |
| 2007 | 0.0706 | 0.0492 | 0.0530 | 0.0290 | 0.0286 | 0.2305 | 30.62% | 21.33% | 23.01% | 12.60% | 12.43% |
| 2008 | 0.0710 | 0.0475 | 0.0513 | 0.0292 | 0.0256 | 0.2245 | 31.64% | 21.17% | 22.83% | 12.99% | 11.38% |
| 2009 | 0.0678 | 0.0449 | 0.0476 | 0.0258 | 0.0227 | 0.2089 | 32.47% | 21.48% | 22.80% | 12.37% | 10.87% |
| 2010 | 0.0616 | 0.0387 | 0.0408 | 0.0152 | 0.0146 | 0.1709 | 36.02% | 22.66% | 23.90% | 8.92% | 8.52% |
| 2011 | 0.0710 | 0.0368 | 0.0494 | 0.0378 | 0.0381 | 0.2331 | 30.46% | 15.80% | 21.17% | 16.23% | 16.33% |
| 2012 | 0.0555 | 0.0413 | 0.0412 | 0.0372 | 0.0324 | 0.2076 | 26.74% | 19.91% | 19.84% | 17.90% | 15.58% |
| 2013 | 0.0527 | 0.0351 | 0.0406 | 0.0315 | 0.0313 | 0.1912 | 27.54% | 18.37% | 21.24% | 16.48% | 16.38% |
| 2014 | 0.0615 | 0.0405 | 0.0419 | 0.0468 | 0.0324 | 0.2231 | 27.55% | 18.17% | 18.78% | 20.97% | 14.54% |
| Mean | 0.0658 | 0.0430 | 0.0478 | 0.0313 | 0.0294 | 0.2174 | 30.37% | 19.86% | 21.98% | 14.35% | 13.43% |

contribution rate of the Midstream displayed a decreasing trend over time, from 10.11% in 2005 to 6.45% in 2014, with an average of 6.80%. Based on the above results, it is concluded that reducing *BGAP* and *WGAP* of the Upstream is the key to the promotion of overall green water use inefficiency in the YREB.

## 4.3 Analysis of the sources of green water use inefficiency

One contribution of this study is to explore the sources of green water use inefficiency in the YREB by decomposing the overall green water use inefficiency into various input-specific components by the EBM model. Then, the contribution rate of each input-specific inefficiency in overall green water use inefficiency is provided. According to this, targeted polices for water-saving and water environmental protection can be formulated by the local governments in the YERB. The input-specific green water use inefficiencies and their contribution rates are presented in Table 5.

It can be seen in Table 5 that, the input-specific green water use inefficiencies of industrial freshwater consumption ($p_1$), agricultural freshwater consumption ($p_2$), domestic freshwater consumption ($p_3$), industrial wastewater emissions ($p_4$) and urban domestic sewage emissions ($p_5$) were 0.0658, 0.0430, 0.0478, 0.0313 and 0.0294 respectively during the sample period. It is found that agricultural water use inefficiency was the primary cause of the overall green water use inefficiency in the YREB, followed by domestic water use inefficiency, industrial water use inefficiency, industrial wastewater emissions inefficiency and urban domestic sewage emissions inefficiency. The contribution rates of the above four input-specific green water use inefficiencies were 30.37%, 21.98%, 19.86%, 14.35% and 13.43%, respectively. This shows that promoting agricultural water use efficiency should be the top priority for the YREB to improve the overall green water use efficiency. From the changing trend perspective, during the sample period, the overall green water use inefficiency in the YREB decreased from 0.2514 to 0.2213. With regard to input-specific green water use inefficiencies, industrial water use inefficiency decreased from 0.0741 to 0.0615, agricultural water use inefficiency decreased from 0.0487 to 0.0405, domestic water use inefficiency decreased from 0.0579 to 0.0419, industrial wastewater emissions inefficiency increased from 0.0320 to 0.0468, and urban domestic sewage emissions inefficiency decreased from 0.0386 to 0.0324, respectively. Furthermore, only the contribution rate of industrial wastewater emissions inefficiency maintained an increasing trend over the sample period, which was from 12.74% to 20.97%, while the contribution rate of the other four input-specific green water use inefficiencies all displayed a decreasing trend.

**Table 6. The potential of water-saving and wastewater emissions in the YREB.**

| Provincial-region | $p_1$ | $p_2$ | $p_3$ | $p_4$ | $p_5$ |
|---|---|---|---|---|---|
| Shanghai | 0.00% | 0.00% | 0.00% | 0.00% | 0.00% |
| Jiangsu | 48.94% | 34.35% | 4.94% | 9.84% | 0.77% |
| Zhejiang | 0.00% | 0.00% | 0.00% | 0.00% | 0.00% |
| Anhui | 84.32% | 47.98% | 42.35% | 25.49% | 19.59% |
| Jiangxi | 85.32% | 50.00% | 51.12% | 29.29% | 26.03% |
| Hubei | 78.76% | 37.61% | 30.99% | 24.58% | 17.85% |
| Hunan | 83.89% | 40.62% | 53.97% | 30.47% | 28.10% |
| Chongqing | 35.72% | 32.66% | 44.26% | 39.03% | 22.87% |
| Sichuan | 44.89% | 9.38% | 27.78% | 9.12% | 14.93% |
| Guizhou | 89.24% | 43.79% | 65.30% | 24.47% | 21.02% |
| Yunnan | 76.09% | 4.99% | 48.88% | 4.81% | 11.72% |
| The Downstream | 16.31% | 11.45% | 1.65% | 3.28% | 0.26% |
| The Midstream | 83.07% | 44.05% | 44.61% | 27.46% | 22.89% |
| The Upstream | 61.49% | 22.70% | 46.55% | 19.36% | 17.63% |
| The YREB | 57.02% | 27.40% | 32.70% | 17.92% | 14.81% |

## 4.4 Analysis of the potential of water-saving and water pollutant emissions

In this section, we estimate the potential of water-saving and water pollutant emissions reduction in the YREB during the sample period according to the difference between each input's actual value and the optimal value, and it is provided in Table 6.

As shown in Table 6, the water-saving potentials of agricultural, industrial and domestic water use in the YREB were respectively 57.02%, 27.40% and 32.70%, and the wastewater emissions reduction potentials of industrial wastewater and urban domestic sewage emissions were 17.92% and 14.81%, respectively. This indicates that there exists serious water waste and water environmental damage in the process of water use in the YREB. At the provincial level, Guizhou, Jiangxi and Anhui had the highest agricultural water-saving potentials of 89.24%, 85.32% and 84.32%, respectively, and the industrial water-saving potential of these three provinces was also in the top three, 43.79%, 50% and 47.98% respectively. Guizhou, Hunan and Jiangxi had the highest domestic water-saving potential of 65.30%, 53.97% and 51.12% respectively. Chongqing, Hunan and Jiangxi had the highest potential for reducing industrial wastewater, reaching 39.03%, 30.47% and 29.29% respectively, and Hunan, Jiangxi and Chongqing had the top three potentials for domestic sewage emissions, which were respectively 28.10%, 26.03% and 22.87%. For the three major areas, it can be inferred that, during the study period, whether it is freshwater-saving potential or wastewater emissions reduction potential, the Midstream was the highest, followed by the Upstream and the lowest Downstream. This indicates that improving green water use efficiency in the Midstream is the key to boosting the overall green water use efficiency and achieving water resources sustainable development for the YREB.

## 4.5 Analysis of the influencing factors of green water use efficiency

To investigate the influencing mechanism of green water use efficiency is considerably helpful for the local governments in the YREB to formulate targeted policies in the future work. Based on this consideration, the main purpose of this section is to test the impact of the influencing factors of green water use efficiency for the overall YREB and its three major sub-regions.

Combining with the existing studies and considering the main focuses of this study, we select the following influencing factors.

(1) Economic development level (*edl*). In our study, GDP per capita is selected as the proxy variable of economic development level for investigating the relationship between the economic development and green water use efficiency in the YREB. Besides, the square term of *edl* is also introduced into the regression analysis for testing the existence of environmental Kuznets curve between green water use efficiency and economic development in the overall region and three sub-regions of YREB.

(2) Water use structure (*wus*). To verify the impact of water use structure on green water use efficiency, the proportion of agricultural freshwater consumption to total regional freshwater consumption is chosen to reflect the water use structure.

(3) Water resources endowment (*wre*). Because water resources endowment denotes the abundance of water resources for a region, we select per capita water resources as a proxy variable for the influencing factor.

(4) Technological innovation (*techi*). Various existing studies have shown that technological innovation is one of the most important ways for freshwater-saving and wastewater emissions reduction, thereby it is beneficial to boosting green water use efficiency. In this paper, the proportion of research and development (R&D) investment to regional GDP is chosen to denote technological innovation level.

(5) Environmental regulation level (*erl*). It is known to us that the government's efforts to strengthen environmental regulation can significantly reduce wastewater emissions, and thus it is conducive to the improvement of green water use efficiency. In this study, the level of regional government's environmental regulation is measured by the ratio of sewage charges over regional GDP.

(6) Regional difference (*rd*). Due to China's uneven regional development, the economic development level, technological level, resources endowment and geographical location of the YREB's three major areas are significantly different. In order to reflect the regional differences influencing green water use efficiency, a dummy variable is introduced. When the dummy variable is 1, it represents the Downstream region, and when it is 0, it represents the Midstream and Upstream region.

The data on these above variables were collected from 'China Statistical Yearbook' (2005–2015), 'China Environmental Statistical Yearbook' (2005–2015) and 'China Science and Technology Statistical Yearbook' (2005–2015). Regarding the selection of regression model, since green water use efficiency is limited dependent variables with value ranging from 0 and 1, therefore a panel Tobit model is employed to estimate the parameters rather than the common-used panel regression approach. Furthermore, a random-effect panel Tobit model is used in our study instead of fixed-effect model. The main reason is that the fixed-effect panel Tobit model cannot provide consistent estimators for regression parameters [51]. Finally, the panel Tobit model for testing the impact of influencing factors on the green water use efficiency in the YREB and its sub-regions is presented as follows.

$$Y_{i,t} = \beta_0 + \beta_1 \ln edl_{i,t} + \beta_2 (\ln edl_{i,t})^2 + \beta_3 wus_{i,t} + \beta_4 wre_{i,t} + \beta_5 techi_{i,t} + \beta_6 erl_{i,t} + \beta_7 rd_{i,t}$$
$$+ \varepsilon_{i,t} \tag{15}$$

Where *Y* is explanatory variable, which refers to the green water use efficiency of the provincial-region of the overall region and its three sub-regions in the YREB; $\beta_i$ ($i$ = 0,1,2,. . .,7) are the coefficients to be estimated; $i$ and $t$ respectively represents region and year; $\varepsilon$ is the stochastic disturbance item and is assumed to be normal with zero mean. It should be pointed out here that there are many factors influencing green water use efficiency and it may be

**Table 7. The regression result of panel Tobit model for the overall region and three sub-regions of the YREB.**

| Independent variables | Overall | $p_1$ | $p_2$ | $p_3$ | $p_4$ | $p_5$ | The Downstream | The Midstream | The Upstream |
|---|---|---|---|---|---|---|---|---|---|
| | Model (1) | Model (2) | Model (3) | Model (4) | Model (5) | Model (6) | Model (7) | Model (8) | Model (9) |
| ln*edl* | 0.4562*** (0.0000) | 0.2180*** (0.0021) | 0.1769*** (0.0000) | 0.1843*** (0.0008) | 0.2147*** (0.0000) | 0.3233*** (0.0045) | 0.7527*** (0.0000) | 0.3380*** (0.0000) | 0.2109*** (0.0000) |
| (ln*edl*)$^2$ | -0.0024*** (0.0076) | -0.0085*** (0.0009) | -0.0009** (0.0187) | -0.0015*** (0.0043) | -0.0018*** (0.0002) | -0.0037*** (0.0024) | -0.0010*** (0.0054) | -0.0034** (0.0130) | -0.0019* (0.0727) |
| *wus* | -0.3490*** (0.0045) | -0.1547*** (0.0032) | -0.2235*** (0.0056) | -0.0984*** (0.0059) | -0.1245** (0.0356) | -0.0781** (0.0121) | -0.6735*** (0.0045) | -0.3211*** (0.0032) | -0.0986*** (0.0056) |
| *wre* | -0.1587*** (0.0022) | -0.1023*** (0.0000) | -0.0890*** (0.0000) | -0.0941** (0.0156) | -0.1712*** (0.0009) | -0.0545*** (0.0052) | -0.0927*** (0.0057) | -0.0323*** (0.0008) | -0.0145*** (0.0018) |
| *techi* | 0.9854*** (0.0000) | 0.4327*** (0.0005) | 1.5217*** (0.0000) | 0.6521*** (0.0000) | 0.9845*** (0.0000) | 0.4215*** (0.0000) | 1.3233*** (0.0000) | 0.8980*** (0.0000) | 0.4539*** (0.0000) |
| *erl* | 0.0898* (0.0741) | -0.1478*** (0.0007) | 0.0096 (0.1436) | -0.1230 (0.1598) | 0.0045** (0.0323) | 0.0547* (0.0562) | 0.1290*** (0.0089) | -0.1517* (0.0745) | 0.0066 (0.2121) |
| *rd* | 0.1247** (0.0254) | 0.2355*** (0.0000) | 0.1479*** (0.0000) | 0.1546** (0.0121) | 0.2574*** (0.0000) | 0.1533** (0.0159) | | | |
| _cons | 0.4570*** (0.0000) | 2.2314*** (0.0000) | 1.2221*** (0.0000) | 0.8954*** (0.0005) | 0.7841 (0.2345) | 0.9292*** (0.0000) | 1.3233*** (0.0000) | 0.5676*** (0.0000) | 0.3459*** (0.0005) |
| Pseudo R$^2$ | 0.2652 | 0.2831 | 0.3033 | 0.1785 | 0.3245 | 0.2895 | 0.3246 | 0.4437 | 0.5055 |
| Log likelihood | -123.5474 | -115.2369 | -98.5621 | -115.3600 | -144.5268 | -132.5470 | -78.3435 | -100.2009 | -112.3401 |

symbols of ***, **, and * respectively denotes 1%, 5%, and 10% significant levels; figures inside the parentheses are p-values.

difficult for us to test all of them. Hence, the selection of influencing factors in this study is mainly based on our research focuses. The regression results are shown in Table 7.

(1) The regression results show that economic development has significantly positive effects on the green water use efficiencies in the overall and three major sub-regions of the YREB and the coefficients of 9 regression models all pass the test at 1% significant level. The main reason for this is that economic development can raise the willingness of freshwater-saving and water environmental protection in society, and the local government also can invest more sufficient funds for water conservancy facilities construction and water pollution control. Besides, it is also found in Table 7 that the regression coefficients of the squared terms of ln*edl* for the overall and five input-specific green water use efficiency are all negative and significant at the 5% level, confirming the existences of environmental Kuznets curve between green water use efficiency and regional economic development level in the YREB. In addition, at the sub-region level, it is observed that the existences of environmental Kuznets curve are confirmed in the Downstream and Midstream, while there is no evidence to support the existence of environmental Kuznets curve in the Upstream.

(2) It is suggested that the proportion of agricultural freshwater consumption to the total freshwater consumption plays significantly negative impacts on green water use efficiencies in the overall region and three sub-regions of the YERB, and that all the coefficients pass the significance test at 5% level. This is mainly because agriculture is China's largest water consumer, which accounts for 65% of China's total freshwater consumption. However, due to the low level of education of China's farmers and the lack of freshwater-saving awareness, serious water resources waste still widely existed during the process of agricultural planting. Moreover, backward agricultural water-saving irrigation technology and inefficient irrigated agricultural systems seriously limits the improvement of China's rural water resources management, thus harmfully ameliorating the green water use efficiency in the YREB.

(3) It can be found that water resources endowment exerts significantly negative effects on green water use efficiencies in the entire region and three sub-regions of the YERB, and that the regression coefficients of the 9 models all pass the significance test at 5% significant level. This indicates that the more water-rich regions, the lower the green water use efficiency. This is mainly because water resources are easily obtained and thus more likely to be wasted in water-rich areas, thereby obstructing the improvement of green water use efficiency. However, water-poor regions may be attracted more attention to water conservation, which is helpful for boosting green water use efficiency.

(4) According to the regression results, increasing R&D investment is beneficial to improving green water use efficiencies in the entire region and three sub-regions of the YERB, and that the regression coefficients of the 9 models all pass the significance test at 1% significant level. It implies that technological innovation plays a particularly important role in freshwater-saving and wastewater emissions control. Thus, improving freshwater-saving and wastewater emissions technologies should be the key measure for freshwater conservation and water environmental protection in the YREB.

(5) It can be inferred that environmental regulation plays significantly positive effects on industrial water use efficiency and industrial wastewater emissions efficiency, and it passes the test of 5% significant level, while the effects on overall green water use efficiency, agricultural water use efficiency, resident water use efficiency and urban domestic sewage emissions efficiency are not significant in the 5% level. A possible explanation of this is that the industrial sector, as the largest polluter in China, is the key sector of Chinese governmental environmental regulation. This is significantly helpful for improving industrial water use efficiency and industrial wastewater emissions efficiency. However, compared with the industrial sector, Chinese governmental environmental regulations imposed on the agricultural sector and residents' living are relatively lower. This could be the main reason for that environmental regulation has no significant impact on the overall and other three input-specific green use efficiencies in the YREB at the 5% level. For the three sub-regions, it is observed that environmental regulation is conducive to improving green water use efficiency in the Downstream and it passes the test at 5% significant level, but its effect on the Midstream and Upstream is not significant without passing the 5% significant level test. This is mainly because environmental regulation in the Downstream is significantly stronger than that in the Midstream and Upstream, and it is helpful for water conservation and water pollution mitigation, indicating that more stringent environmental regulation should be imposed on the Midstream and Upstream for the improvement of green water use efficiency.

(6) It is shown that the coefficients of the dummy variables are all positive and pass the test of 5% significant level, indicating that the regional differences play significant impacts on green water use efficiency in the YREB. The main reason is that compared with the other two major areas in the YREB, the Downstream enjoys more superior geographical location, higher economic openness and economic development level, sufficient funds, developed technology and higher human capital level, which is more conducive to promoting green water use efficiency.

## 5. Conclusions

In recent years, the issues of water resources conservation and water environmental protection have received more attentions. Improving green water use efficiency is the only way to walk out of the current water resources and environmental dilemma. This study tries to analyze green water use efficiency of the YREB during the period of 2005–2014. To do this, a new DEA approach namely EBM model was first proposed for measuring green water use efficiency, and

the Theil index is introduced to explore the sources of regional gap in the YREB. Then, a random-effect panel Tobit model is conducted for testing the impact of the influencing factors on the overall green water use efficiency and input-specific green water use efficiencies in the YREB. Conclusions and the corresponding implications are as follows:

(1) Green water use efficiency in the YREB is relatively low, and there is still a large potential for freshwater-saving and wastewater emissions reduction. The relatively lower green water use efficiency suggests that there is considerable potential for improvement. In addition, there are significant regional differences in green water use efficiency among YREB's different provincial-regions and three major areas. It is found that green water use efficiency gaps between three major areas and the provincial green water use efficiency gaps within the Upstream is the main sources for the gaps among the YREB's provincial-regions. Narrowing the green water use efficiency gaps between different regions/areas is one of the most important way for improving green water use efficiency. It calls for the joint effort for every region/area in the YREB. All regions/areas in the YREB should carefully analyze their freshwater consumption and wastewater emissions characteristics, and then take more targeted water-saving strategies and wastewater reduction measures. Furthermore, all regions/areas in the YREB should avoid local protection and expand exchanges and cooperation, and then share advanced freshwater saving and wastewater reduction technologies with each other.

(2) The contribution rates of agricultural freshwater use, industrial freshwater use, domestic freshwater use, industrial wastewater emissions and urban domestic sewage emissions to the overall green water use inefficiency in the YREB were respectively 30.37%, 19.86%, 21.98%, 14.35% and 13.43%. Agricultural, industrial and domestic freshwater use respectively had 57.02%, 27.40% and 32.70% freshwater-saving potential, and industrial wastewater and urban domestic sewage had 17.92% and 14.81% reduction potential respectively. Therefore, to improve the overall green water use efficiency, all regions/areas in the YREB should raise the agricultural freshwater use efficiency as the focus of future green water use efficiency promotion. Concretely, accelerating farmland water conservancy construction, raising agricultural environmental regulation level, adjusting agricultural irrigation water prices, actively promoting agricultural water-saving irrigation technology, and formulating a stricter agricultural water management system are all important ways for realizing the potential of agricultural water-saving in the YREB.

(3) Economic development, technological innovation, and regional differences significantly promoted the overall green water use efficiency and input-specific green water use efficiencies in the YREB, while the increase of agricultural water use proportion and the water resources endowment significantly inhibited the improvement of the overall green water use efficiency and input-specific green water use efficiencies in the YREB. Environmental regulation played a significantly positive effect on improving industrial water use efficiency and industrial wastewater emissions efficiency, but has no significant effects on agricultural water use efficiency, domestic water use efficiency and urban domestic sewage emissions efficiency. In the future work, in order to improve green water use efficiency, all regions/areas in the YREB should eliminate their backward high-freshwater consumption technologies, improve freshwater-saving and recycling technologies, and boost their water resources management level at the same time. A more stringent water resources management system and a scientific water rights trading system also should be established as quickly as possible. Moreover, all regions/areas in the YREB should increase R&D investment for freshwater-saving and water pollution control technologies, relying on technological innovation to promote green water use efficiency. Governmental environmental regulation should be further boosted and more stringent measures should be established in the overall YREB, especially in the Midstream and Upstream. Besides this, formulating reasonable urban sewage emissions standards, improving the supervision

mechanism for urban domestic sewage emissions, and redesigning the current sewage charge system are all important works for improving green water use efficiency in the future for the local government in the YREB.

## Supporting information

**S1 Data.**
(XLSX)

## Author Contributions

**Conceptualization:** Mingsong Zhao.

**Formal analysis:** Lili Ding.

**Funding acquisition:** Jianming Wang.

**Supervision:** Qunwei Wang.

**Writing – original draft:** Jianguo Wang.

**Writing – review & editing:** Ke-Liang Wang.

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
