## [Decision Letter · Decision Letter 0]

26 Nov 2019

PONE-D-19-28340

Investigating the spatiotemporal differences and influencing factors of green water use efficiency of Yangtze River Economic Belt in China

PLOS ONE

Dear Dr. WANG,

Thank you for submitting your manuscript to PLOS ONE. After careful consideration, we feel that it has merit but does not fully meet PLOS ONE’s publication criteria as it currently stands. Therefore, we invite you to submit a revised version of the manuscript that addresses the points raised during the review process.

We would appreciate receiving your revised manuscript by Jan 10 2020 11:59PM. To enhance the reproducibility of your results, we recommend that if applicable you deposit your laboratory protocols in protocols.io, where a protocol can be assigned its own identifier (DOI) such that it can be cited independently in the future. For instructions see: http://journals.plos.org/plosone/s/submission-guidelines#loc-laboratory-protocols

We look forward to receiving your revised manuscript.

Kind regards,

Bing Xue, Ph.D.

Academic Editor

PLOS ONE

Journal Requirements:

Reviewers' comments:

Reviewer's Responses to Questions

**Comments to the Author**

1. Is the manuscript technically sound, and do the data support the conclusions?

Reviewer #1: Yes

Reviewer #2: Partly

2. Has the statistical analysis been performed appropriately and rigorously? 

Reviewer #1: Yes

Reviewer #2: No

3. Have the authors made all data underlying the findings in their manuscript fully available?

Reviewer #1: No

Reviewer #2: Yes

4. Is the manuscript presented in an intelligible fashion and written in standard English?

Reviewer #1: Yes

Reviewer #2: No

5. Review Comments to the Author

Reviewer #1: In this manuscript, the authors assess the green water use efficiency (GWUE) of Yangtze River Economic Belt (YREB) by the epsilon-based measure model in DEA and investigate the spatial differences and influencing factors of the overall GWUE and input-specific GWUE. The paper is well written and I would like to suggest minor revision with the following comments and suggestions:

• The introduction section could be improved by highlighting the theoretical contributions to the literature. The current narratives tend to emphasize the practical meaning for the YREB and the methodological debate and contributions. It is suggested to add a transitional paragraph about theories or empirics of water efficiency in subjects like agricultural, industrial, and domestic usage before diving into the methodological discussion of DEA.

• The pattern analysis in the results section is good but the regression analysis could be further improved. Since the modeling involves the longitudinal data from 2005 to 2014, the fixed effects and random effects need to be dealt with, which implies the use of time dummy or mixed effects model technique. The economic development level (edl) variable could be further extended to include both the edl and edl^2 to test the environmental Kuznets curve for the overall GWUE or input-specific GWUE.

• The text in section 4 is inconsistent with section 3. In section 4, it is implied that p1 variable is agricultural water use inefficiency, e.g. “(In table 5) With regard to input-specific green water use inefficiencies, agricultural water use inefficiency decreased from 0.0741 to 0.0615”. But in section 3, p1 is industrial water consumption.

• Since YREB is a regional planning concept, it is needed to introduce the concept and relevant planning document from the State Council or National Development and Reform Commission in the introduction or section 3.

• I think “as shown in Table 2, Fig. 2 and Fig. 3” in page 10 actually means ““as shown in Table 3, Fig. 2 and Fig. 3”.

• The manuscript should be further checked regarding the language and reference styles. There are typos or errors like: “it need not obtain” in page 3, “spa differences” in page 4, “four input-specific” in page 15, year “2108” in ref. of Yang, Wang, and Geng.

Reviewer #2: This paper has evaluated the spatiotemporal differences of green water use efficiency (GWUE) in the Yangtze River Economic Belt (YREB) and calculated the contribution rate of each input-specific green water use inefficiency in the overall green water use efficiency and the potential of water-saving and water pollution reduction. However, this manuscript has a number of issues which need to be explained further, or to be clarified clearly.

My specific comments are given as follows:

1. The definition of green water use efficiency in this paper is not clear. Specifically, the authors defined that green water use efficiency (GWUE) as a ratio between economic value added and environmental pressures. For environmental pressure, it was integrated by a single indicator taking freshwater consumption and waste water into account. However, what about their weightings? Which one may contribute larger pressure to environment? Besides, in actual industrial production, the reclaimed water has been significantly reused to mitigate freshwater consumption and waste water emissions. The authors have not considered such features to adjust the indicator.

2. In the part of background, there is a lack of holistic literature review regarding comparison of the existing studies to highlight the contribution.

3. The authors address that the epsilon-based measure (EBM) model is a reasonable way to overcome the weakness of the radial and non-radial models. There is a lack of the comparison to show its advantage and potentials.

4. In the “Methodology” part, the actual indication of each variable should be given under each formula.

5. The authors indicated that the green water use efficiency of the Yangtze River Economic Belt performed spatial difference. However, there are a number of factors to give rise in such difference. Especially, each factor may be significantly different in various regions. The study just applied the global regression analysis to the entire region of Yangtze River Economic Belt, how could it lead to resources management recommendations for different regions are challenged.

6. In the part of “Empirical results and discussion”, there are no strong evidences to support the author's points. For instance, “According to the original statistical data, it can be found that the current urban domestic sewage emissions has to some extent exceeded the industrial wastewater emissions in China”.

7. The English writing needs to be thoroughly improved. The jargons should be used instead of the gossips. For instance, water resources consumption and water pollution emission should be replaced by freshwater consumption and waste water emissions.

6. PLOS authors have the option to publish the peer review history of their article (what does this mean?). If published, this will include your full peer review and any attached files.

Reviewer #1: No

Reviewer #2: No

---

## [Author Response · Author response to Decision Letter 0]

8 Jan 2020

Reviewer #1:

Major comments:

In this manuscript, the authors assess the green water use efficiency (GWUE) of Yangtze River Economic Belt (YREB) by the epsilon-based measure model in DEA and investigate the spatial differences and influencing factors of the overall GWUE and input-specific GWUE. The paper is well written and I would like to suggest minor revision with the following comments and suggestions:

1. The introduction section could be improved by highlighting the theoretical contributions to the literature. The current narratives tend to emphasize the practical meaning for the YREB and the methodological debate and contributions. It is suggested to add a transitional paragraph about theories or empirics of water efficiency in subjects like agricultural, industrial, and domestic usage before diving into the methodological discussion of DEA.

Response:

Thanks very much for the suggestion. We have added a paragraph to describe the green water use efficiency theoretically before the discussion of DEA in the revised manuscript. The added paragraph is as follows.

‘Water use efficiency, defined by the ratio of the optimal water input to the actual water input of an economic unit, is an important indicator for measuring the level of water resource utilization for a country, region or firm. However, most of the existing water use efficiency related studies mainly focused on the water conservation but did not consider the discharge of wastewater emissions, and thus the results were biased. To address this problem, this paper incorporates wastewater emissions into the total-factor water use efficiency measurement framework and defines a new water use efficiency indicator named “green water use efficiency (GWUE)” by utilizing a novel DEA model— EBM model. The new indicator takes into account both freshwater conservation and wastewater mitigation simultaneously, thereby providing a more scientific and comprehensive approach for water use efficiency measurement.’

2. The pattern analysis in the results section is good but the regression analysis could be further improved. Since the modeling involves the longitudinal data from 2005 to 2014, the fixed effects and random effects need to be dealt with, which implies the use of time dummy or mixed effects model technique. The economic development level (edl) variable could be further extended to include both the edl and edl^2 to test the environmental Kuznets curve for the overall GWUE or input-specific GWUE.

Response:

Thanks very much for your constructive suggestions.

Firstly, reviewer pointed out that fixed effect and random effect should be considered in the regression analysis, and we strongly agree with this suggestion. Considering the explanatory variable is limited dependent variable with its value ranging from 0 to 1, we used the panel Tobit model to estimate the parameters rather than the common panel regression model. Moreover, we use a random-effect Tobit model instead of a fixed-effect Tobit model. The reasons are mainly twofold. On the one hand, for the fixed-effect Tobit model, since the sufficient statistics for individual heterogeneity cannot be found, conditional maximum likelihood estimation cannot be performed like the fixed-effect Logit model. On the other hand, if dummy variables (similar to LSDV) are directly added to the pooled Tobit model, the fixed effect estimates obtained are also inconsistent. Therefore, we only considered to utilize the random-effect Tobit model. In the revised manuscript, we have added related description for the selection of regression model, which is as follows.

‘Regarding the selection of regression model, since green water use efficiency is limited dependent variables with value ranging from 0 and 1, therefore a panel Tobit model is employed to estimate the parameters rather than the common-used panel regression approach. Furthermore, a random-effect panel Tobit model is used in our study instead of fixed-effect model. The main reason is that the fixed-effect panel Tobit model cannot provide consistent estimators for regression parameters (MaDonald and Moffitt, 1980). Finally, the panel Tobit model for testing the impact of influencing factors on the green water use efficiency in the YREB and its sub-regions is presented as follows.’

Secondly, according to reviewer’s suggestion, both the edl and edl^2 are introduced into the regress equations and the existences of environmental Kuznets curve are tested, and the parameters of other explanatory variables were re-estimated meantime. The estimation results confirmed the existences of the environmental Kuznets curve for the overall GWUE and input-specific GWUEs. 

In the revised manuscript, the result of testing the environmental Kuznets curve has been added.

‘Besides, it is also found in Table 7 that the regression coefficients of the squared terms of lnedl for the overall and five input-specific green water use efficiency are all negative and significant at the 5% level, confirming the existences of environmental Kuznets curve between green water use efficiency and regional economic development level in the YREB. In addition, at the sub-region level, it is observed that the existences of environmental Kuznets curve are confirmed in the Downstream and Midstream, while there is no evidence to support the existence of environmental Kuznets curve in the Upstream.’

3. The text in section 4 is inconsistent with section 3. In section 4, it is implied that p1 variable is agricultural water use inefficiency, e.g. “(In table 5) With regard to input-specific green water use inefficiencies, agricultural water use inefficiency decreased from 0.0741 to 0.0615”. But in section 3, p1 is industrial water consumption.

Response:

Thank you very much for pointing out this error. We have corrected it in the revised manuscript.

4. Since YREB is a regional planning concept, it is needed to introduce the concept and relevant planning document from the State Council or National Development and Reform Commission in the introduction or section 3.

Response:

Thank you very much for your suggestion. We have added a description related to China’s Yangtze River Economic Belt strategy in the revised manuscript, which is shown in the following.

“The Yangtze River Economic Belt (YREB)” was first proposed in the 1980s, which originated from the “Yangtze River Industrial Dense Belt” proposed by the China Productivity Economics Research Association, which is an economic belt along Yangtze River with Shanghai as the leader. Promoting the development of the YREB has become a national strategy in 2015.

5. I think “as shown in Table 2, Fig. 2 and Fig. 3” in page 10 actually means ““as shown in Table 3, Fig. 2 and Fig. 3”.

Response:

Thank you very much for pointing out this error. We have corrected it in the revised manuscript.

6. The manuscript should be further checked regarding the language and reference styles. There are typos or errors like: “it need not obtain” in page 3, “spa differences” in page 4, “four input-specific” in page 15, year “2108” in ref. of Yang, Wang, and Geng.

Response:

Thank you very for pointing out these errors. We have carefully checked the manuscript and corrected these errors in the revised manuscript.

Reviewer #2: 

Major comments:

This paper has evaluated the spatiotemporal differences of green water use efficiency (GWUE) in the Yangtze River Economic Belt (YREB) and calculated the contribution rate of each input-specific green water use inefficiency in the overall green water use efficiency and the potential of water-saving and water pollution reduction. However, this manuscript has a number of issues which need to be explained further, or to be clarified clearly.

My specific comments are given as follows:

1. The definition of green water use efficiency in this paper is not clear. Specifically, the authors defined that green water use efficiency (GWUE) as a ratio between economic value added and environmental pressures. For environmental pressure, it was integrated by a single indicator taking freshwater consumption and waste water into account. However, what about their weightings? Which one may contribute larger pressure to environment? Besides, in actual industrial production, the reclaimed water has been significantly reused to mitigate freshwater consumption and waste water emissions. The authors have not considered such features to adjust the indicator.

Response:

**Thanks for very much for your constructive comment. Please allow us explain the definition of green water use efficiency and identification of the weights of water environmental pressures.

In this paper, we define green water use efficiency as a ratio between economic value added and water environmental pressures with freshwater consumption and wastewater emissions being considered as water environmental pressures. Similar to the definition of eco-efficiency proposed by Kuosmanen and Kortelainen (2005), green water use efficiency is defined as follows in our study.

Green water use efficiency=v/P(pi) (i=1,2,…,m)

Where v represents the economic value generated in the production process, P is the pressure function that aggregates the m water environmental pressures into a single water environmental pressure score. Because value added can be either directly obtained or indirectly computed using data on prices and quantities of outputs and intermediate inputs, constructing the composite water environmental pressure score is a difficult problem. Kuosmanen and Kortelainen (2005) proposed that a reasonable approach to computing this score is to take a weighted average of particular pressures on the environment. In our study, employing the approach proposed by Kuosmanen and Kortelainen (2005), we construct an aggregated water environmental pressure indicator P(pi), which is as follows.

P(pi)=w1p1+w2p2+….+wmpm (i=1,2,…,m)

Where wi (i=1,…,m) is the weight with which pressure i enters into the computation of the water environmental pressure score.

Therefore, constructing composite indictor of water environmental pressure P requires different weights to be assigned to different water environmental pressure pi (i=1,…,n), and the weight reflects the relative importance of the different water environmental pressure pi (i=1,…,n). In order to avoid the bias stemming from a subjective choice of common weights, in this paper we have decided to use DEA as aggregation method, and the weight of each water environmental pressure is inherently determined by the DEA model, which avoids the subjective bias and thereby improve the objectivity and accuracy of green water use efficiency evaluation.

**Thank you very much for your suggestion that reclaimed water should be considered in the framework of green water use efficiency evaluation. Undoubtedly, this is a better choice, and we strongly agree your suggestion. However, due to that the data of provincial reclaimed water cannot directly be obtained in China, it is regrettable that we did not introduce the reclaimed water into our study. Of course, your suggestion is an important extension in our future research work. Thank you very much!

2. In the part of background, there is a lack of holistic literature review regarding comparison of the existing studies to highlight the contribution.

Response:

Thank you very much for pointing out the defect in this paper. According to your suggestion, some holistic literature review regarding comparison of the existing studies and the major contribution have been added in the revised manuscript, which is as follows.

‘Li and Ma (2014) used SBM-undesirable and meta-frontier models to evaluate China’s industrial water use efficiency and investigated the impact factors. Wang et al. (2015) explored agricultural water-use efficiency in the Heihe River Basin in the Northwest China with DEA and Malmquist productivity index. Deng et al. (2016) employed SBM-DEA model to evaluate water use efficiency of 31 provinces in China with the sewage as undesirable output. Zhao et al. (2017) adopted two-stage SBM model to estimate water resource utilization efficiency with taking environmental constraint into account and investigated the spatial spillover effect. Song et al. (2018) applied Malmquist-Luenberger productivity index to measure provincial water resource efficiency in China and used panel Tobit model to explore the factors that affect water resource efficiency. Yao et al. (2018) measured green total factor water efficiency by using SBM model considering undesirable output. Hu et al. (2018) utilized a super efficiency DEA to evaluate the water use and wastewater treatment efficiency of 10 cities in the Minjiang River Basin of China. In these studies, the economic outputs as well as negative externalities generated (e.g. water pollution emissions) during water use activity are well considered. Therefore, water use efficiency evaluations which consider water pollution emissions can provide more reasonable and more accurate estimation results.’

‘In recent years, the EBM model has been widely applied in the field of evaluation of efficiency and productivity. Qin et al. (2017) employed global EBM model to estimate the energy efficiency with air emission as undesirable output in China’s coastal areas. Xu and Cui (2017) applied a new approach combining network EBM model and network SBM model to evaluate the overall energy efficiency and divisional efficiency of 19 international airlines. Cui and Li (2017) used a dynamic EBM model to evaluate the dynamic efficiency of 19 international airlines. The above studies show that the EBM model is feasible and effective for measuring the efficiency of product units with the advantages of the radial and non-radial models simultaneously. However, to the best of our knowledge, studies that use the EBM model to measure water use efficiency have not been found. To fill this gap, in this paper, we utilize a window-based EBM model to investigate the water use efficiency of the YERB at a more detailed level. Major contributions of this study can be summarized as follows. First, it combines the EBM model and window-based DEA approach to evaluate and decompose provincial green water use efficiency in the YREB. Second, the Theil index is employed to identify the sources of the gap of green water use efficiency among the three major areas of the YREB. Third, a random-effect panel Tobit model is used to test the influencing factors of green water use efficiency in the YREB.’

3. The authors address that the epsilon-based measure (EBM) model is a reasonable way to overcome the weakness of the radial and non-radial models. There is a lack of the comparison to show its advantage and potentials.

Response:

Thank you very much for pointing out this defect in our study. The advantage of EBM model is that it enjoys the advantages of the radial and non-radial models and overcomes the weaknesses of the two type of models simultaneously. In recent years, the EBM has been widely applied in the field of efficiency measurement. In the revised manuscript, some representative studies such as Qin et al. (2017), Cui and Li (2017), Xu and Cui (2017) have been added in the part of background, which are as follows.

‘In recent years, the EBM model has been widely applied in the field of evaluation of efficiency and productivity. Qin et al. (2017) employed global EBM model to estimate the energy efficiency with air emission as undesirable output in China’s coastal areas. Xu and Cui (2017) applied a new approach combining network EBM model and network SBM model to evaluate the overall energy efficiency and divisional efficiency of 19 international airlines. Cui and Li (2017) used a dynamic EBM model to evaluate the dynamic efficiency of 19 international airlines. The above studies show that the EBM model is feasible and effective for measuring the efficiency of product units with the advantages of the radial and non-radial models simultaneously.’

4. In the “Methodology” part, the actual indication of each variable should be given under each formula.

Response:

Thank you very much for your suggestion. According to your suggestion, we have carefully checked each formula and ensured to give the actual indication of each variable in the revised manuscript.

5. The authors indicated that the green water use efficiency of the Yangtze River Economic Belt performed spatial difference. However, there are a number of factors to give rise in such difference. Especially, each factor may be significantly different in various regions. The study just applied the global regression analysis to the entire region of Yangtze River Economic Belt, how could it lead to resources management recommendations for different regions are challenged.

Response:

Thank you very much for your thoughtful comment. As you said, since the Yangtze River Economic Belt embodies a vast territory and has unbalanced economic development level, the results show that significant spatial differences exist among different sub-regions in terms of green water use efficiency. Due to different national and socio-economic elements, the influencing factors of green water use efficiency in the three major sub-regions of Yangtze River Economic Belt are also different. Therefore, according your suggestion, for the purpose of investigating the regional differences of influencing factors, three regression equations (model7, model8, model9) with the green water use efficiency of three major sub-regions as explanatory variables have been added and more detailed information related to improving the green water use efficiency for different sub-regions in the Yangtze River Economic Belt also can be obtained. The regression results are shown in page 20 in the revised manuscript.

6. In the part of “Empirical results and discussion”, there are no strong evidences to support the author's points. For instance, “According to the original statistical data, it can be found that the current urban domestic sewage emissions have to some extent exceeded the industrial wastewater emissions in China”.

Response:

Thank you very much for your comment. We have checked carefully and corrected related description in the part of “Empirical results and discussion” for the purpose of providing stronger evidences to support our points in the revised manuscript. 

7. The English writing needs to be thoroughly improved. The jargons should be used instead of the gossips. For instance, water resources consumption and water pollution emission should be replaced by freshwater consumption and waste water emissions.

Response:

Thank you very much for pointing out this defect in our manuscript. According to your suggestion, we have checked carefully and corrected the English writing in the revised manuscript.

In addition, due to the contribution during the process of our manuscript revision, Dr. Lili Ding has been added as a co-author in the revised manuscript.

---

## [Decision Letter · Decision Letter 1]

13 Mar 2020

Investigating the spatiotemporal differences and influencing factors of green water use efficiency of Yangtze River Economic Belt in China

PONE-D-19-28340R1

Dear Dr. WANG,

We are pleased to inform you that your manuscript has been judged scientifically suitable for publication and will be formally accepted for publication once it complies with all outstanding technical requirements.

With kind regards,

Bing Xue, Ph.D.

Academic Editor

PLOS ONE

Additional Editor Comments (optional):

Reviewers' comments:

Reviewer's Responses to Questions

**Comments to the Author**

1. If the authors have adequately addressed your comments raised in a previous round of review and you feel that this manuscript is now acceptable for publication, you may indicate that here to bypass the “Comments to the Author” section, enter your conflict of interest statement in the “Confidential to Editor” section, and submit your "Accept" recommendation.

Reviewer #1: All comments have been addressed

2. Is the manuscript technically sound, and do the data support the conclusions?

Reviewer #1: Yes

3. Has the statistical analysis been performed appropriately and rigorously? 

Reviewer #1: Yes

4. Have the authors made all data underlying the findings in their manuscript fully available?

Reviewer #1: Yes

5. Is the manuscript presented in an intelligible fashion and written in standard English?

Reviewer #1: Yes

6. Review Comments to the Author

Reviewer #1: In the revision, the authors have dealt with the comments and suggestions in a satisfying way. Specifically, the contributions of the manuscript are highlighted, and the methods and results are improved by introducing the fixed-effect panel Tobit model and the validation of environmental Kuznets hypothesis. I would like to suggest accepting the revision for the publication of PONE journal.

7. PLOS authors have the option to publish the peer review history of their article (what does this mean?). If published, this will include your full peer review and any attached files.

Reviewer #1: No

---

## [Editor Report · Acceptance letter]

16 Mar 2020

PONE-D-19-28340R1 

Investigating the spatiotemporal differences and influencing factors of green water use efficiency of Yangtze River Economic Belt in China 

Dear Dr. WANG:

I am pleased to inform you that your manuscript has been deemed suitable for publication in PLOS ONE. Congratulations! Your manuscript is now with our production department. 

With kind regards,

on behalf of

Professor Bing Xue 

Academic Editor

PLOS ONE